# The relationships of spiritual health, pregnancy worries and stress and perceived social support with childbirth fear and experience: A path analysis

Saeideh Hosaini[1], Mansoureh Yazdkhasti[2], Farnoosh Moafi Ghafari[3], Farima Mohamadi[4], Seyed Hamid Reza Kamran Rad [5], Zohreh Mahmoodi [2]*

1 Social Determinants of Health Research Institute for Prevention of Noncommiiunicable Disease, Qazvin University of medical sciences, Qazvin, Iran, 2 Social Determinants of Health Research Center, Alborz University of Medical Sciences, Karaj, Iran, 3 Department of Midwifery, School of Nursing and Midwifery, Qazvin University of Medical Sciences, Qazvin, Iran, 4 Social Determinants of Health Research Center, Shahid Beheshti University of Medical Sciences, Tehran, Iran, 5 Faculty of Education and Psychology, Shahid Beheshti University, Tehran, Iran

* zohrehmahmoodi2011@gmail.com

## Abstract

### Background

Given maternal health is a major health indicator, the present research aimed at determining the causal relationships of spiritual health, worries, stress and perceived social support with the fear and experience of childbirth in pregnant women.

### Methods

The present longitudinal prospective research recruited 352 pregnant women presenting to selected health centers in Qazvin, Iran in 2021. The data were collected using the Childbirth Experience Questionnaire-2 (CEQ-2), the Wijma Delivery Expectancy/Experience Questionnaire (W-DEQ), the Multidimensional Scale of Perceived Social Support (MSPSS), the Persian version of the Pregnancy Worries and Stress Questionnaire (PWSQ), the Spiritual Health Questionnaire, the Socioeconomic Status (SES) questionnaire and a sociodemographic checklist, and were analyzed in SPSS-25 and Lisrel-8.8.

### Results

The mean age of the participants was 28.1±6.8 years. According to the results of the path analysis, among the variables related to fear of childbirth, childbirth experience (B = -0.37, CI:-0.44;-0.22) in the direct path and perceived social support (B = -0.51, CI:-0.58;-0.43) in both direct and indirect paths demonstrated the most significant negative relationship. Among the variables related to childbirth experience, pregnancy worries and stress had a negative causal relationship (B = -0.06, CI:-0.079;-0.043) in the direct path, spiritual health showed the highest significant positive relationship (B = 0.01, CI: 0.008; 0.012) in the indirect path, and perceived social support (B = 0.112, CI: 0.092; 0.131) and the number of

**Data Availability Statement:** working link to dataset are as follows: TARGET URL: (https://doi.

org/10.5281/zenodo.10183716) https://zenodo.
org/records/10183716.

**Funding:** The author(s) received no specific
funding for this work.

**Competing interests:** The authors have declared
that no competing interests exist.

**Abbreviations:** SES, Socio-Economic Statues; GR,
Gravid; CN, child number; EDUM, Education
mother; EDUH, Education Husband; SP, Spiritual
well-being; SS, Social support; FE, Fear of
childbirth; TEN, Pregnancy's Worries and Stress;
EX, Childbirth experience.

children (B = 0.32,CI: 0.30; 0.34) demonstrated the highest significant positive relationship in both direct and indirect paths. In other words, childbirth experience becomes more desirable as spiritual health, social support, and the number of children increases, and it becomes less desirable as pregnancy worries and stress rise.

## Conclusion

According to the present findings, various psychological, social, and spiritual factors are associated with childbirth fear and experience. It is thus necessary to utilize appropriate methods and promote training and support to reduce the adverse outcomes of childbirth.

## 1. Background

The maternal mortality rate is one of the key health indicators of any country directly or indirectly affected by pregnancy and childbirth [1] Moreover, these two experiences constitute the most important events in a woman's life [2], as a mother's experiences of pregnancy and childbirth can have desirable or undesirable short- or long-term effects on her own, her family's, and the newborn's life. The most commonly-reported health concerns in pregnancy relate to childbirth, neonatal health and parenting [3,4].

Childbirth is a process that is not wholly predetermined, and its outcomes cannot be predicted. It is multidimensional and can include all kinds of feelings from happiness and satisfaction to anxiety. The differences are related to mothers perceived of the situations, culture, and religion, emotional well-being and staffs behavior. Uncertainty about the birth process seems to be a reason for fear of childbirth [5]. A quarter of mothers experience different levels of childbirth fear due to their fear of episiotomy, losing control, and pain [6], which can affect the mother and the newborn [7,8]. These two concepts are so closely related that high fear of childbirth can lead to an unpleasant childbirth experience for the mother. Conversely, an unpleasant previous childbirth experience can cause higher fear in the mother in the recent birth [2,9,10]. According to the literature, the prevalence of undesirable experiences and fear of childbirth varies in different countries and cultures [11]. For instance, 19.8% of the mother assessed in a study in Turkey and 6.1% of those in a study in Iran experienced a severe fear of childbirth [4,12].

Numerous factors, such as demographic, psychosocial, and spiritual characteristics, are associated with adverse experiences and fear of childbirth. Some researchers found that psychological and social factors have an effective role in causing these issues [13]. Also, according to the Fear-Tension-Pain theory, fear of childbirth, maternal tension, and the amount of pain experienced are cyclically related and can affect each [14].

In a conceptual model developed in 2015, Siddall stated the role of spiritual, physiological, social, and psychological factors on labor pain. According to this model, pain is a multidimensional concept, and modifying or reinforcing each of its dimensions can change the mother's perception of it. For instance, mothers' social support and higher spiritual health lead to better tolerance of labor pain, followed by less fear and a better experience of childbirth (Fig 1) [15].

In addition, spiritual health and perceived social support can help as a coping mechanism to control oneself in difficult and stressful situations and show better adaptation [16,17]. Although the results of various studies show the positive effect of spirituality on mental health, we should consider that religious beliefs may lead to negative outcomes by encouraging people to quit treatment or delaying theirtimely referral to prevent diseases [18]. For example, in a

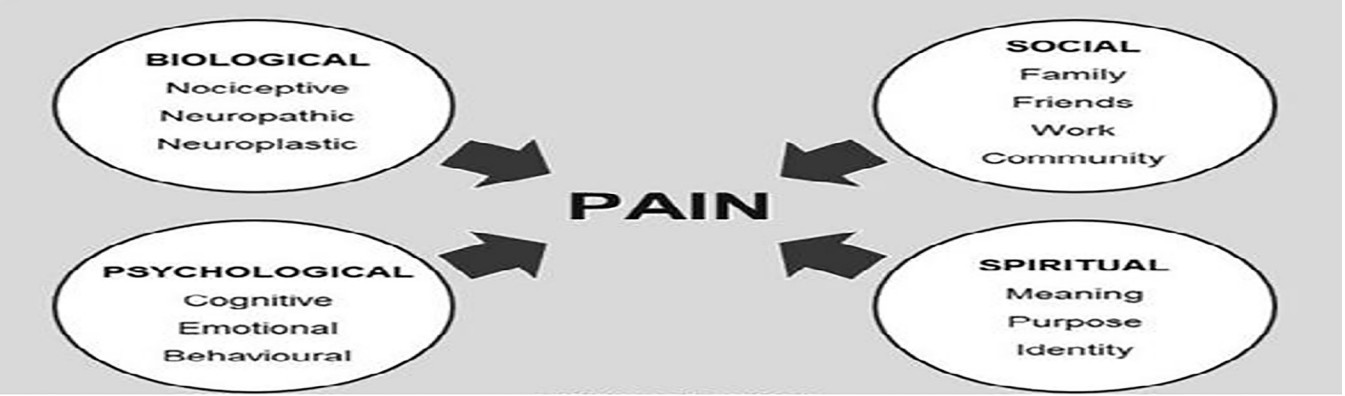

**Fig 1. The model presented by Siddall based on the role of spiritual, physiological, social and psychological factors in labor pain in pregnant mother** [15].

study by Beery et al. with 250 patients in England, they found that patients with stronger spiritual beliefs had a worse prognosis and worse condition than other patients during nine months of continuous follow-up [19].

To the best of the authors' knowledge, perceived social support, spiritual and psychological variables, and fear and experience of childbirth have not been addressed yet in a single model. The present research thus aimed at determining the causal associations of spiritual health, worries, stress and perceived social support with the experience and fear of childbirth in pregnant women.

The questions raised were as follows:

1. What is the effect of spiritual well-being, perceived social support and pregnancy worries and stress (direct/indirect) on fear of childbirth in pregnant mothers?

2. What is the effect of spiritual well-being, perceived social support and pregnancy worries and stress (direct/indirect) on childbirth experience in pregnant mothers?

3. What is the childbirth experience (direct/indirect) on fear of childbirth in pregnant mothers?

4. What is the effect of demographic factors (age, education) on pregnancy worries and stress, fear of childbirth and childbirth experience in pregnant mothers?

5. Do pregnancy worries and stress mediate the effect of fear of childbirth on childbirth experience?

## 2. Methods

### 2.1. Study design and participants

This longitudinal (prospective) study was performed in 2021 on 352 eligible pregnant mothers presenting to the selected health centers in Qazvin, Iran. Qazvin is the largest city and the capital of Qazvin Province in the central part of Iran.

Based on the previous study results [20], the minimum required sample size considering the first type error 0.05 for a two-way test ($\alpha = 0.05$), the second type error 0.2 ($\beta = 0.2$) (test power 0.8) and considering the correlation coefficient between the social support and fear and experience of childbirth, at least 0.16 and using the below formula; the minimum sample size

was 302 mothers. Therefore, the final sample size was 352 mothers considering 15% loss.

$$n = \frac{\left(z_{1-\frac{\alpha}{2}} + z_{(1-\beta)}\right)^2}{r^2} + 3$$

### Inclusion criteria

Iranian pregnant mothers who were in the last four weeks of their pregnancy; not having high-risk pregnancies, such as multiple pregnancies, preeclampsia, and gestational diabetes; not having a mental illness according to the self-reports or health records, and not taking antidepressants and anti-anxiety medications according to the self-reports or health records, not having negative birth and abortion experiences and pregnancies were planned and desired.

### Exclusion criteria

Not having a phone number and lack of access to the parturient to complete the questionnaires in the second stage of the study; delivery in a center other than the selected centers; returning incomplete questionnaires, and withdrawing from the study.

### 2.2. Data collection and definition of terms

Data were collected by a sociodemographic checklist and the Wijma Delivery Expectancy/ Experience Questionnaire (W-DEQ), Childbirth Experience Questionnaire 2 (CEQ-2), Multidimensional Scale of Perceived Social Support (MSPSS), Spiritual Health Questionnaire, the Persian version of the Pregnancy Worries and Stress Questionnaire (PWSQ), and Ghodrat-nama Socioeconomic Status (SES) questionnaire.

**2.2.1. Demographic checklist.** This checklist included items on the couple's age, education, gravidity, the number of children, and Pregnancy-Childbirth History.

**2.2.2. Wijma Delivery Expectancy/Experience Questionnaire (W-DEQ).** Wijma K. et al. designed a specific questionnaire known as the Wijma Delivery Expectancy/Experience Questionnaire (W-DEQ) to assess fear of childbirth [21]. In 2017, Mortazavi translated the W-DEQ for use in Iran and examined its validity and reliability. They reported a reliability of 0.91 for this scale [20]. The W-DEQ has 33 items, which are scored based on a six-point Likert scale from 0 to 5. The total score for W-DEQ is between 0 and 165. Higher scores denote greater fear of childbirth in the mothers. A score above 100 indicates severe fear in the mother [22].

**2.2.3. Childbirth Experience Questionnaire 2 (CEQ-2).** The participating mother's childbirth experience was examined using a specific questionnaire, namely the Childbirth Experience Questionnaire 2 (CEQ-2). The CEQ-2 was developed by Dencker et al. [23]. In Iran, midwifery researchers translated this tool into Persian, validated it in 2020 and reported a reliability of 0.91, indicating that the tool is suitable for being implemented in Iranian society. CEQ-2 has 23 items divided into four domains: own capacity, professional support, perceived safety, and participation. Twenty of its items are scored based on a four-point Likert scale from strongly agree (4 points) to strongly disagree (1 point). Three items are scored based on a ruler from 0 to 100. The score of these three items is also converted to a score from 1 to 4 (i.e., 0–40 = 1 point, 41–60 = 2 points, 61–80 = 3 points, 80–100 = 4 points). A higher final score indicates the mother's more positive experience, and a lower score shows a more negative experience [24].

**2.2.4. MSPSS.**    The 12-item MSPSS developed by Zimet et al [25]. Helps measure the support provided by the friends family and significant others on a seven-point Likert scale, ranging from "strongly disagree" to "strongly agree". In 2013, Bagherian et al. confirmed the reliability and validity of this scale by calculating a Cronbach's alpha of 0.84 [26]. Similarly, the present research confirmed the reliability by obtaining a Cronbach's alpha of 0.82.

**2.2.5. Spiritual health questionnaire.**    The 20-item Spiritual Health Questionnaire (Palutzian & Ellison) was used to evaluate spiritual health [27]. Ten items measured existential health and ten items measured religious health on a scale of 10–60. Given the lack of definite religious and existential health subgroups, judgments were made on the basis of the scores obtained. The higher the score, the better the religious and existential health. The spiritual health score was obtained as 20–120, i.e. the sum of the scores of these two subscales. The spiritual health therefore positively related to the total score of this tool. The items were scored on a six-point Likert scale, ranging from "strongly disagree" to "strongly agree". Rezaei et al. confirmed the reliability and validity of this instrument by calculating a reliability coefficient of 0.79 [28].

**2.2.6. PWSQ.**    The 25-item PWSQ constitutes a combination of the 10-item scale designed by Hoysing et al. [29] and certain personal/family factors addressed in the original version of the Van den Berg questionnaire [30]. This questionnaire comprises six subscales, i.e. maternal health (6 items), neonatal health (5 items), mother-newborn bonding (2 items), experience of childbirth and motherhood (4 items), personal-occupational (3 items) and personal-family (5 items). This tool was scored on a five-point Likert scale defined as 0: never, 1: rarely, 2: sometimes, 3: often and 4: always, with a total score of 0–100. Despite the lack of cut-off points in the PWSQ, the score showed the level of worry and its effective factors in pregnancy. Identifying these factors and offering solutions can help lower worries and anxiety in pregnant women and prevent the harmful effects of stress. The validity of the PWSQ was confirmed by Navidpour et al. (2015) in Iran using the face, content and construct validity. The criterion validity assessments also showed significant correlations between this questionnaire and the Spiel Berger State-Trait Anxiety Inventory (r = 0.739, P<0.001) [31].

**2.2.7. Socioeconomic status questionnaire.**    Four dimensions of socioeconomic status, i.e. housing status, income level, education and economic class, were evaluated using the socioeconomic status questionnaire (Ghodratnama, 2013) consisting of five main items and six demographic items. The items were scored on a five-point scale ranging from 1: very low to 5: very high. In Iran, Eslami et al. approved the face and content validity of this questionnaire and confirmed its reliability through calculating a Cronbach's alpha of 0.83 [32].

## 2.3. Procedures

The present research began after obtaining the necessary permission and approval of the Ethics Committee of Alborz University of Medical Sciences, Karaj, Iran. After presenting to the health centers, the researcher identified the eligible individuals, briefed them on the study objectives and asked them to sign informed consent forms for participation in the study.

Data were collected in two stages in this study from 2021/3/3 to 2021/9/2.

The first stage was performed during the last four weeks when the mothers visited the selected centers for periodic examinations. Information related to their spiritual health, pregnancy worries and stress, and fear of childbirth were collected during these visits. After completing the questionnaires, the mothers gave the researcher their telephone numbers to contact after their delivery. If the mother could not complete the questionnaires in one meeting, the researcher would set the next time to complete the questionnaires

The second stage was performed after the delivery. At this stage, one week before the estimated date of delivery, the mother was contacted via a phone call and asked to inform the

researcher when they attended the hospital. The researcher then visited the hospital to collect information on the mother's childbirth experience and perceived social support when she went to the postpartum ward and was in a more stable condition. If a mother's clinical condition were not suitable for any reason, the completion of the questionnaires would be postponed until the mother's condition stabilized. If the mother gave birth earlier than the scheduled date, which was based on the date of the first day of her last menstruation, she was contacted, and arrangements were made for her to complete the delivery experience and perceived support questionnaires at the time of her subsequent referral to the select center for postpartum care (i.e., 3 to 5 days after the delivery).

Ethical approval and consent to participate

Informed consent was obtained from all the participants. All the methods were employed based on the relevant guidelines and regulations. The Ethics Committee of Alborz University of Medical Sciences approved all the experimental protocols (IR. ABZUMSREC.1399.273).

After briefing the eligible candidates on the study objectives, they signed written informed consent forms. They were assured of the confidentiality of their information and their right to withdraw from the study at their own discretion without being deprived of health services.

## 2.4. Statistical analysis

According to Fig 2, the present study investigated the fit of a conceptual model of the relationships of spiritual health, worries, stress and perceived social support with the fear and experience of childbirth in pregnant women. The Kolmogorov-Smirnov test was employed to examine the distribution normality of the quantitative data. The path analysis was performed as an extension of conventional regression to show both direct and indirect effects of the individual variables on the dependent variables. The results of this analysis were used to interpret the relationships and correlations. The data were analyzed in SPSS-25 [33] and Lisrel-8.8 [34]. The Pearson's correlation coefficient was also used to express the correlations and the beta coefficient to report the path analysis. The level of statistical significance was adjusted at $T > 1.96$.

## 3. Results

The present study examined the data from 352 pregnant mother presenting to the selected centers in Qazvin, Iran. The mean age of the mother was 28.1±6.8 years and that of their husbands 33.6±6.2. Most mother were housewives, and most of their husbands were self-employed (95.5%). The mean score of spiritual health was 103.12±15.8, pregnancy worries and stress 37.2±20.8, perceived social support 66.3±15.2, fear of childbirth 59.6±26.8, and childbirth experience 61.2±10.4 (Table 1).

According to the results of Pearson's correlation test, the score of childbirth experience had a positive correlation with the number of children, spiritual health score, and perceived social support and a negative correlation with mother's education, fear of childbirth, and pregnancy worries and stress. Among those variables, the number of children had the highest positive correlation (r = 0.149) and fear of childbirth with the highest negative correlation (r = -0.459) with childbirth experience. In other words, the chances of a desirable childbirth experience decreased as fear of childbirth grew (Table 2).

According to the findings, female education and the number of children had a positive correlation, and socioeconomic status, social support and spiritual health had a negative and significant correlation with fear of childbirth, among which the number of children had the highest positive correlation (r = 0.182) and spiritual health had the highest inverse correlation.

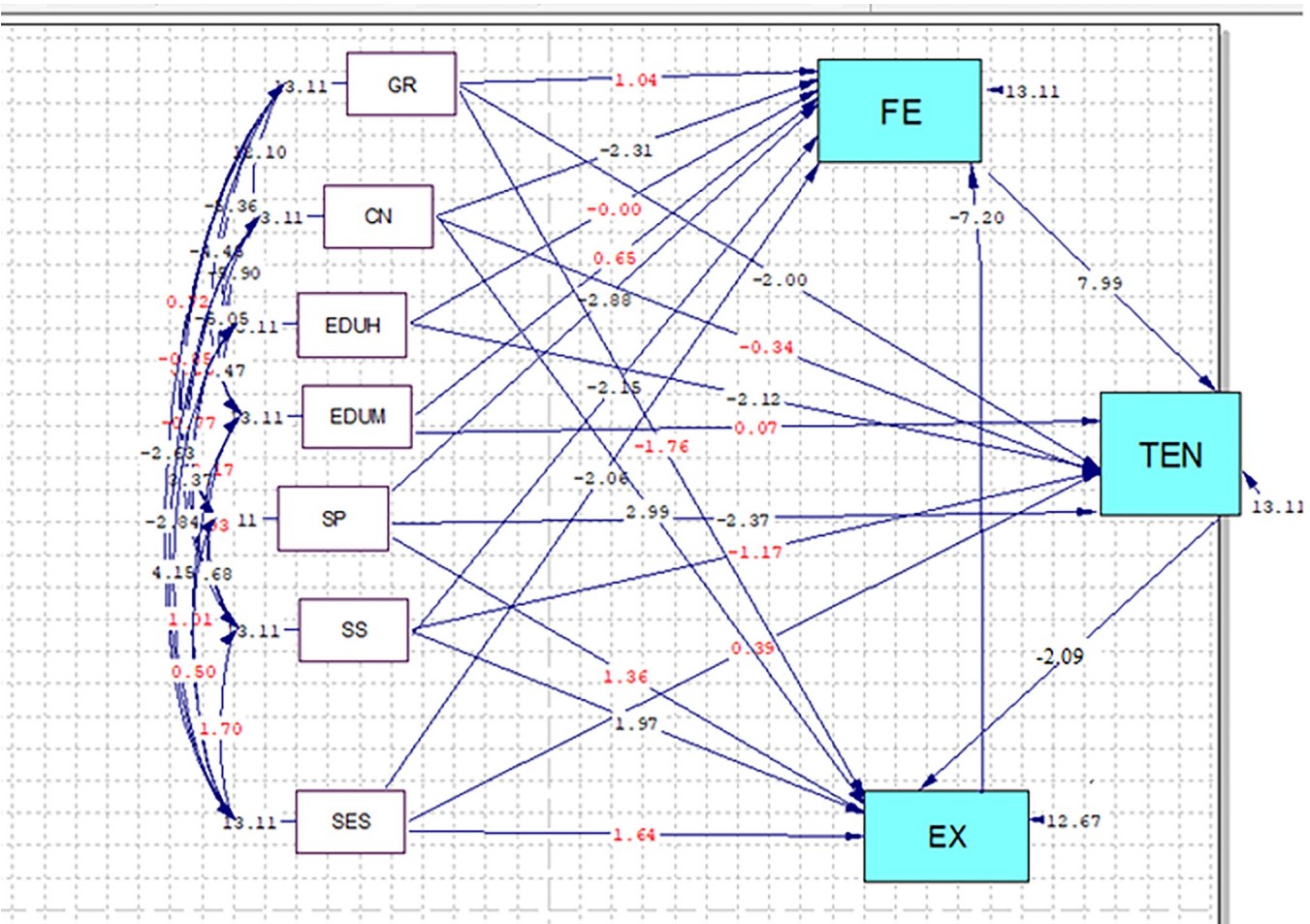

**Fig 2. Full empirical path model between spiritual health, pregnancy worries and stress, and perceived social support with childbirth fear and experience according T-Value ≥1.96.** Red number isn't significant. SES = Socio-Economic Statues, GR = Gravid, CN = child number, EDUM = Education mother EDUH = Education Husband, SP = Spiritual well-being, SS = Social support, FE = Fear of child birth, TEN = Pregnancy's Worries and Stress, EX = Childbirth experience.

Furthermore, with the fear of childbirth (r = -0.200), In other words, increasing the score of spiritual health is associated with reducing fear of childbirth.

Based on the results of the path analysis, after examining the paths that were significant due to a T-value ≥1.96 (Fig 2), the variables of spiritual health (B = -0.14), childbirth experience (B = -0.37), and socioeconomic status (B = -0.1) in the direct path, and pregnancy worries and stress (B = 0.022) in the indirect path, were associated with fear of childbirth, while perceived social support (B = -0.51) and the number of children (B = -0.334) had a significant negative causal relationship with fear of childbirth in both direct and indirect paths. In other words, the score of fear of childbirth decreases as the score of the noted variables increases.

Based on the findings, the variable of pregnancy worries and stress had a negative and significant causal relationship (B = -0.06) with childbirth experience in the direct path; in the indirect path, the variable of husband's education (B = 0.007), gravidity (B = 0.01), socioeconomic status (B = 0.002), and spiritual health (B = 0.01) had a positive and fear of childbirth (B = -0.02) has negative and significant causal relationship with childbirth experience; meanwhile, the variables of perceived social support (B = 0.112) and the number of children (B = 0.32) had

**Table 1. The sociodemographic character of participants.**

| quantitative | | | |
|---|---|---|---|
| Variables | Mean ± sd | Variables | Mean ± sd |
| Age mother (year) | 8/1±6/28 | Spiritual health | 103.12 ±15.8 |
| Age men | 2/6±6/33 | Perceived social support | 66.3±15.2 |
| Education mother(year) | 9.8±3.5 | Fear of birth | 59.6±26.8 |
| Pregnancy's Worries and Stress | 37.2±20.8 | Childbirth experience | 10.4 61.2± |
| Education men (year) | 9.4±3.7 | Gestational age | 2/0±2/38 |
| Socioeconomic status | 6/2±3/12 | | |

| qualitative | | | | | |
|---|---|---|---|---|---|
| Variables | | F (%) | Variables | | F (%) |
| Number of Family | <2 | 163(57) | JOB mother | House keeper | 336(95.5) |
| | ≥2 | 119(41.6) | | Worker | 7(2) |
| Number of children | Zero | 132(37.5) | | Employment | 9(2.6) |
| | 1 | 124(35.2) | | | |
| | 2 | 67(19) | | | |
| | 3 and more | 29(8.3) | | | |

a positive and significant causal relationship with childbirth experience in both direct and indirect paths. That is to say; the childbirth experience becomes more desirable as spiritual health, socioeconomic status, social support, and the number of children increase. In contrast, it becomes less desirable as pregnancy worries and stress rise. According these finding pregnancy worries and stress was mediator between fear of child birth and childbirth experience (Fig 3) (Table 3).

The results of the model fit indices indicated the desirability and high fit of the model and the rationality of the adjusted relationships between the variables based on the conceptual model. Accordingly, the fitted model does not differ significantly from the conceptual model (Table 4).

## 4. Discussion

Pregnancy and childbirth are events that change a woman's worldview and affect her health, emotions, and social roles [35]. Fear is a normal emotion that can be an appropriate response to danger or threat [36].

**Table 2. The correlation matrix of childbirth fear and experience in relation to personal-social variables, spiritual health, pregnancy worries and stress, and perceived social support.**

| | | 1 | 2 | 3 | 4 | 5 | 6 | 7 | 8 | 9 | 10 |
|---|---|---|---|---|---|---|---|---|---|---|---|
| 1 | Mother's education | 1 | | | | | | | | | |
| 2 | Husband's education | 0.513* | 1 | | | | | | | | |
| 3 | Gravidity | -0.247** | -0.302** | 1 | | | | | | | |
| 4 | Number of children | -0.283** | -0.335** | 0.861** | 1 | | | | | | |
| 5 | Socio-economic status score | 0.055 | 0.230 | -0.143 | -0.155** | 1 | | | | | |
| 6 | Spiritual health score | 0.009 | 0.025 | 0.039 | 0.014 | 0.027 | 1 | | | | |
| 7 | Perceived social support score | 0.105* | 0.185** | -0.046 | -0.042 | 0.092 | 0.203** | 1 | | | |
| 8 | Fear of childbirth score | 0.105* | 0.053 | -0.118* | 0.182** | -0.120* | -0.200** | -0.183** | 1 | | |
| 9 | Pregnancy worries and stress score | 0.034 | -0.039 | -0.225** | -0.225** | -0.036 | -0.216** | -0.165** | 0.484** | 1 | |
| 10 | Childbirth experience score | -0.133* | -0.097 | 0.84 | 0.149** | 0.078 | 0.108* | 0.134* | -0.459** | -0.272** | 1 |

* P<0.05

** P<0.01.

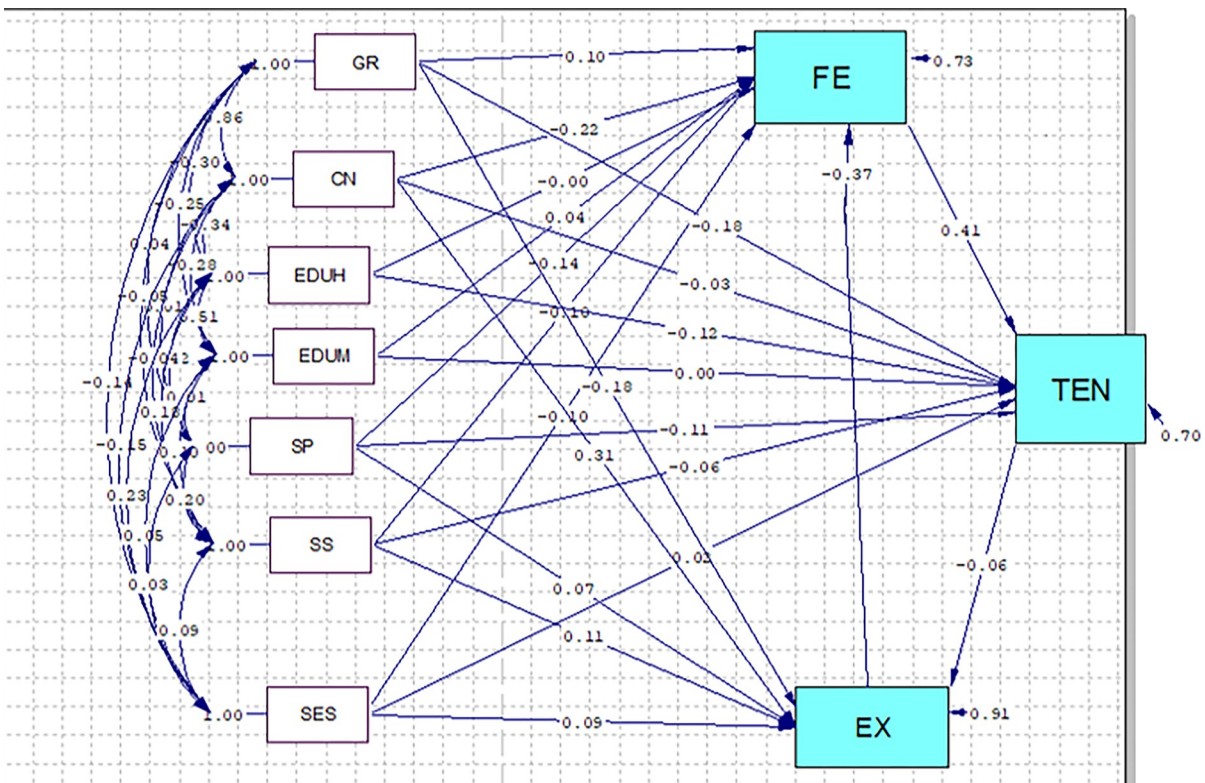

**Fig 3. Full Empirical Path Model between spiritual health, pregnancy worries and stress, and perceived social support with childbirth fear and experience.** Single-headed arrow means regression coefficient, Standardized Beta. SES = Socio-Economic Statues, GR = Gravid, CN = child number, EDUM = Education mother EDUH = Education Husband, SP = Spiritual well-being, SS = Social support, FE = Fear of childbirth, TEN = Pregnancy's Worries and Stress, EX = Childbirth experience.

The path analysis revealed the most significant and negative causal relationship between childbirth experience and fear of childbirth. Research suggests an unpleasant history of childbirth raises the fear of the following childbirth in the mother [2,9,10]. When someone experiences an unpleasant event, this memory or experience is stored in a different brain region as primary sensory memory. Therefore, it can lead to later understandings of disturbing images and thoughts [37]. In their qualitative study, Rodríguez-Almagro et al. (2019) found that childbirth complications, obstetric problems, and prenatal problems can affect a mother's childbirth experience [38]. Korukcu et al. (2017) and Viirman et all (2022) also found that fear of childbirth results from a previous unfavorable pregnancy and childbirth experience and Primary contributing factors to negative childbirth experiences appear to be labour- and birth-related [39,40].

Perceived social support was also mostly associated with the fear of childbirth along both direct and indirect paths. Social support functions as an intermediary between stress and its manifestations. This variable is negatively related to anxiety and positively to adaptation [17]. According to Dolatian et al. (2014), perceived social support affects gestational age and prevents preterm delivery by influencing worries and stress in pregnancy [41]. Research suggests the key role of perceived support provided by the husband, family and friends in lowering the fear of childbirth [12,42,43].

Based on the present study findings, pregnancy worries and stress had the most destructive relationship with childbirth experience in the direct path and fear of childbirth in indirect

**Table 3. The direct and indirect effects of personal-social variables, spiritual health, pregnancy worries and stress, and perceived social support on childbirth fear and experience.**

| Variables | | Direct effects | Indirect effects | Total effect | R² |
|---|---|---|---|---|---|
| **Fear of childbirth** | Mother's education | 0.04 | - | 0.04 | 0.30 |
| | Husband's education | 0.00 | - | 0.00 | |
| | Gravidity | 0.1 | 0.066 | 0.166 | |
| | Number of children | -0.22* | -0.114* | -0.334* | |
| | Socio-economic status | -0.1* | -0.033 | -0.1* | |
| | Perceived social support | -0.1* | -0.041* | -0.51* | |
| | Spiritual health | -0.14* | 0.0259 | -0.14* | |
| | Childbirth experience | -0.37* | - | -0.37* | |
| | Pregnancy worries and stress | - | 0.022* | 0.022* | |
| **Childbirth experience** | Mother's education | - | -0.001 | -0.001 | 0.40 |
| | Husband's education | - | 0.007* | 0.007* | |
| | Gravidity | -0.18 | 0.01* | 0.01* | |
| | Number of children | 0.31* | 0.01* | 0.32* | |
| | Socio-economic status | 0.09 | 0.002* | 0.002* | |
| | Perceived social support | 0.11* | 0.002* | 0.112* | |
| | Spiritual health | 0.07 | 0.01* | 0.01* | |
| | Fear of childbirth | - | -0.02* | -0.02* | |
| | Pregnancy worries and stress | -0.06* | - | -0.06* | |

path. In other word this variable was mediator between fear of childbirth and childbirth experience. Studies have shown a correlation between stress and stressful events during pregnancy and adverse pregnancy outcomes [13,38,44]. Stress can affect the childbirth process and lead to adverse pregnancy outcomes such as preterm delivery through the hypothalamic-pituitary-endocrine axis. In this way, it is related to an undesirable pregnancy experience [45].

Spiritual health was another variable that was indirectly, positively, and significantly related to pregnancy experience. Spirituality is the most critical guide in problem-solving behaviors. Studies have shown that people with high levels of spirituality cope better with problems and life circumstances. Spiritual health helps reduce fear of childbirth and leads to a more pleasant pregnancy and childbirth experience by controlling stress and promoting psychosocial health [46]. Bilgiç et al. (2021) found a negative correlation between spiritual health and fear of childbirth [4]. Foruzandeh Hafshejani et al. (2018) found a linear relationship between spiritual health and stress coping styles [47].

Perceived social support and the number of children had a positive and significant relationship with pregnancy experience in both paths. As stated earlier, perceived social support significantly affects fear, adverse pregnancy outcomes, and pregnancy experience by controlling worries and stress [43,48].

The number of children was another variable that was positively correlated with pregnancy experience. In nulliparous mothers, fear of childbirth may be due to the lack of information and receiving incorrect information from others. Nonetheless, some studies have reported the

**Table 4. Goodness of fit indices for the model.**

| Fitting Index | X² | df | X²/df | CFI | GFI | NFI | RMSEA |
|---|---|---|---|---|---|---|---|
| Model Index | 91/5 | 2 | 9/2 | 1 | 1 | 1 | 03/0 |
| Acceptable Range | X2/df < 5 | | | > 0.9 | > 0.9 | > 0.9 | < 0.05 |

NFI = Normed-fit index, GFI = Goodness-of-fit statistic, RMSEA = Root mean square error of approximation, $X^2$ = chi-square.

relationship of multi-parity with fear of birth and poor pregnancy experience [39]. This disparity can also be due to the differences in the cultural context and, subsequently, the spiritual health of individuals [46]. Moreover, some studies have suggested a relationship between having previous childbirth experience and increased self-confidence in the mother. This history can significantly impact the mother's experience during the next pregnancies [49].

## 5. Limitation

One of the limitations of the present research is that we used questionnaires to collect and record the data, and the number of questions can affect individual's accuracy. Furthermore, we assessed pregnant mothers presenting to selected health centers, not the ones who went to other private centers. Also, some women did not present to health care centers during the COVID-19 pandemic due to social distancing rules. These could have affected our sampling; therefore, now that the conditions are changed and more mothers can be examined, it is suggested that more extensive research be conducted even in private centers.

## 6. Conclusion

Our study showed that spiritual health, and perceived social support is related to childbirth fear and experience in pregnant mothers. Accordingly, using appropriate programs to improve spiritual and family support can reduce fear and adverse outcomes of childbirth and make positive childbirth experience. Other findings were the relationship between childbirth experience, fear of childbirth and the media role of pregnancy worries and stress so health systems must have attention to these subjects to prepare programs to aware mothers about pregnancy and delivery for decreasing their fear and pregnancy worries and stress, also prepare programs for health workers to make well childbirth experience for mothers.

## Supporting information

**S1 Checklist. STROBE statement—checklist of items that should be included in reports of observational studies.**
(DOCX)

## Acknowledgments

The present research was extracted from a master's dissertation on midwifery counselling. The authors would like to express their gratitude to the participants and the authorities of the Research Deputy and Education Deputy of Alborz University of Medical Sciences for their support.

## Author Contributions

**Conceptualization:** Zohreh Mahmoodi.

**Data curation:** Saeideh Hosaini, Farnoosh Moafi Ghafari.

**Formal analysis:** Farima Mohamadi, Zohreh Mahmoodi.

**Investigation:** Mansoureh Yazdkhasti.

**Methodology:** Farima Mohamadi, Zohreh Mahmoodi.

**Software:** Farnoosh Moafi Ghafari.

**Validation:** Mansoureh Yazdkhasti, Zohreh Mahmoodi.

**Writing – original draft:** Saeideh Hosaini, Mansoureh Yazdkhasti, Farnoosh Moafi Ghafari, Farima Mohamadi, Zohreh Mahmoodi.

**Writing – review & editing:** Saeideh Hosaini, Mansoureh Yazdkhasti, Farnoosh Moafi Ghafari, Farima Mohamadi, Seyed Hamid Reza Kamran Rad, Zohreh Mahmoodi.

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
