## [Decision Letter · Decision Letter 0]

20 Jun 2023

PONE-D-23-08753The relationship of spiritual health, pregnancy worries and stress, and perceived social support with childbirth fear and experience: A path analysisPLOS ONE

Dear Dr. Zohreh Mahmoodi,

Thank you for submitting your manuscript to PLOS ONE. After careful consideration, we feel that it has merit but does not fully meet PLOS ONE’s publication criteria as it currently stands. Therefore, we invite you to submit a revised version of the manuscript that addresses the points raised during the review process.

We look forward to receiving your revised manuscript.

Kind regards,

Zemenu Yohannes Kassa, Msc

Academic Editor

PLOS ONE

Journal Requirements:

2. We noticed you have some minor occurrence of overlapping text with the following previous publication, which needs to be addressed:

Shadabi, N., Saeieh, S. E., Qorbani, M., Babaheidari, T. B., & Mahmoodi, Z. (2021). The relationship of supportive roles with mental health and satisfaction with life in female household heads in Karaj, Iran: a structural equations model. BMC public health, 21, 1-9.

In your revision ensure you cite all your sources (including your own works), and quote or rephrase any duplicated text outside the methods section. Further consideration is dependent on these concerns being addressed.

Additional Editor Comments from academic editor:

# Abstract: Your only show us the β -value. However, you should show us a confidence interval.

Did you conduct a multivariable analysis? if yes, which domains show statistically significant? If not, why?

Conclusion and recommendations

The conclusion and recommendations are given based on your pertinent findings while you recommended physical aspects, there are no findings in your result part related to physical domains.

#Introduction

You should use the words mother or women across the document to clarify the reader.

In Paragraph 2, you expected to demonstrate the women’s fear during childbirth and its outcome, which makes the reader more interested in your ideas follow.

You did not clearly explain the research gaps rather, listing research questions. You should remove research questions and add what makes your findings differ from the existing articles.

#Result

The mean score of spiritual health was 104.3±14.9 out of ________?

Reviewers' comments:

Reviewer's Responses to Questions

**Comments to the Author**

1. Is the manuscript technically sound, and do the data support the conclusions?

Reviewer #1: Partly

Reviewer #2: Partly

2. Has the statistical analysis been performed appropriately and rigorously? 

Reviewer #1: I Don't Know

Reviewer #2: No

3. Have the authors made all data underlying the findings in their manuscript fully available?

Reviewer #1: No

Reviewer #2: Yes

4. Is the manuscript presented in an intelligible fashion and written in standard English?

Reviewer #1: No

Reviewer #2: No

5. Review Comments to the Author

Reviewer #1: The relationship of spiritual health, pregnancy worries and stress, and perceived social support with childbirth fear and experience: A path analysis

1. What about related studies? In addition, what is gap of the previous studies in these issues?

2. Why researchers used general social support scale for pregnant women? The study needs specific scale in this issue.

3. Considering the questionnaire length in this study and in pregnant women? What about arrangements in the study to prevent the accuracy decrease?

4. Why researchers used online consent form?

5. "Assuming the correlation between the social support and fear and experience of childbirth, at least 0.16"author express reference?

6. Persian Version of the Pregnancy Worries and Stress Questionnaire (PWSQ)

7. This tools generally was used in assessing anxiety ?

8. "The second stage was performed after the delivery. At this stage, one week before the probable date of the delivery, the mother was contacted via a phone call and asked to inform the researcher when they attended the hospital. The researcher then visited the hospital to collect information on the mother's childbirth experience and perceived social support when she went to the postpartum ward and was in a more stable condition. If a mother's clinical condition were not suitable for any reason, the completion of the questionnaires would be postponed until the mother's condition stabilized". If women did 9.Which kind of social support (instrumental support-emotional – information- appraisal) had the highest negative causal relationship with fear in pregnant women?

Reviewer #2: 1. The first aim of the current study is to examine the effect of spiritual well-being, perceived social support and pregnancy's worries and stress on fear of childbirth. However, little information about pregnancy's worries and stress can be found in the Introduction part. Furthermore, what kind of variable is pregnancy's worries and stress? An independent variable of fear of childbirth? Or a mediating variable between fear of childbirth and childbirth experience? In Figure 2, the path analysis revealed a potential mediation model among PE, TEN, and EX. Therefore, from my humble point, the results of the study do not fit the aims. The authors should explain clearly the relationships of the five variables (spiritual well-being, perceived social support, pregnancy's worries and stress, fear of childbirth, and childbirth experience) in the Introduction part.

2. From the Introduction part, perceived social support might be an independent variable of fear of childbirth. However, in Method part, fear of childbirth was measured in the first stage, while perceived social support was measured in the second stage. It is very confusing and the authors should explain it.

3. The minimum required sample size (352) is equal to actual sample size (352)?

4. There are some obvious limitations in the study. However, the Limitation part is short and non-professional. The authors should re-write this part.

5. The Conclusion part is less related to the findings of the study. For example, countries have always considered mothers' health as one of the most important groups in society. One of the most important health indicators of a country is its maternal mortality rate. Is it the conclusion from the results of the current study? The authors should re-write this part and conclude what we can get from the findings of the current study.

6. In Table 3, pregnancy worries and stress was included in the model of childbirth experience but excluded in the model of fear of childbirth. It is not consistent with the aims of the study. The authors should explain the reason.

7. The format of tables is not standard, such as Table 1. Moreover, the format of fonts is inconsistent, such as Table 1 and Table 2. The authors should uniform it.

6. PLOS authors have the option to publish the peer review history of their article (what does this mean?). If published, this will include your full peer review and any attached files.

Reviewer #1: No

Reviewer #2: No

---

## [Author Response · Author response to Decision Letter 0]

3 Jul 2023

Dear Editor 

Thank you for your valuable comments. We corrected and answered all of them as follows:

Journal Requirements:

1. Please ensure that your manuscript meets PLOS ONE's style requirements, including those for file naming

Answer: Thank you. We checked it.

2. We noticed you have some minor occurrence of overlapping text with the following previous publication, which needs to be addressed:

Shadabi, N., Saeieh, S. E., Qorbani, M., Babaheidari, T. B., & Mahmoodi, Z. (2021). The relationship of supportive roles with mental health and satisfaction with life in female household heads in Karaj, Iran: a structural equations model. BMC public health, 21, 1-9.

Answer: Thank you. We checked and corrected it.

3.Please provide additional details regarding participant consent. In the ethics statement in the Methods and online submission information, please ensure that you have specified (1) whether consent was informed and (2) what type you obtained (for instance, written or verbal, and if verbal, how it was documented and witnessed). If your study included minors, state whether you obtained consent from parents or guardians. If the need for consent was waived by the ethics committee, please include this information.

Answer: Thank you for your attention. The eligible people explained the study objectives to them and asked them to read and sign a written informed consent form if they wished to participate in the study. This is explained in the ‘Ethics approval and consent to participate’ section.

Additional Editor Comments from academic editor: 

Abstract: Your only show us the β -value. However, you should show us a confidence interval.

Answer: Thank you for your attention. It was added as per your comments.

Did you conduct a multivariable analysis? If yes, which domains show statistically significant? If not, why?

Answer: Thank you for your attention. Yes, path analysis is an extension of multiple regression that allows us to examine more complicated relations among the variables. It is used to answer research questions about the effect of a given independent (X1) variable on the dependent variable (Y) in the model. (1) In this study, we assessed the direct/ indirect effects of some independent variables: mother’s education, father’s education, gravidity, number of children, socioeconomic statues, perceived social support, spiritual health and pregnancy worries and stress on fear of childbirth and childbirth experience. In the analysis, we used the total number of questionnaires (socioeconomic statues, perceived social support spiritual health and pregnancy worries and stress) not their domains. And as we wrote in Methods, the paths are significant if T ≥ 1.96 (Figur-2). The paths coefficients -direct /indirect- are shown in Table 3 and Figur-3.

Conclusion and recommendations

the conclusion and recommendations are given based on your pertinent findings while you recommended physical aspects, there are no findings in your result part related to physical domains.

Answer: Thank you for your attention. Sorry for the mistake. We corrected it.

#Introduction

you should use the words mother or women across the document to clarify the reader.

Answer: Thank you for your attention. We corrected it. We replaced women with mothers.

In Paragraph 2, you expected to demonstrate the women’s fear during childbirth and its outcome, which makes the reader more interested in your ideas follow.

You did not clearly explain the research gaps rather, listing research questions. You should remove research questions and add what makes your findings differ from the existing articles.

Answer: Thank you for your attention. It was added this in two paragraphs.

#Result

the mean score of spiritual health was 104.3±14.9 out of ________?

Answer: Thank you for your attention. It is 103.12 ±15.8. Sorry for the typo. As we explained in Methods section, spiritual health total score lies between 20-120. Most participants scored higher than 90.

Reviewer 1:

1. What about related studies? In addition, what is gap of the previous studies in these issues?

Answer: Thank you for your comments. We added some studies to the Introduction section as per reviewer comments. For further explanation, we did not find the models that assess all of these factors together with fear and childbirth experience. As explained, we wrote a conceptual model developed in 2015 about labor pain. So it needs to assess the factors that are effective on these variables to find the suitable approach. 

2. Why researchers used general social support scale for pregnant women? The study needs specific scale in this issue.

Answer: Thank you for your comments. In our study, we did not use general social support (Vaux et al., 1986). We assessed the Perceived Social Support (Zimet et al.). The Multidimensional Scale of Perceived Social Support (MSPSS) is a short and reliable instrument that assesses perceived social support from the social network of an individual (family, friends and significant others). This questionnaire was assessed in pregnant women too and its reliability and validity during pregnancy in pregnant women were confirmed: “Psychometric Validation of the Multidimensional Scale of Perceived Social Support during Pregnancy in Rural Pakistan. 2021 (2)

3. Considering the questionnaire length in this study and in pregnant women? What about arrangements in the study to prevent the accuracy decrease?

Answer: Thank you for your attention. As written in the Method section in 2.3. Procedures, questionnaires were collected in two phases:

The first phase was during the last four weeks of pregnancy when the spiritual health, pregnancy worries and stress, and fear of childbirth questionnaire were collected. If a mother could not complete the questionnaires in one meeting, the researcher would set the next time to complete the questionnaires. 

The second phase was after delivery. The childbirth experience and perceived social support were collected when the mother went to the postpartum ward and was in a more stable condition. If a mother's clinical condition was not suitable for any reason, the completion of the questionnaires would be postponed until the mother's condition stabilized.

We explained this more in the Methods section. But it is our limitation and we mentioned it in the limitation section.

4. Why researchers used online consent form?

Answer: Thank you for your comments. We did not use online consent form. As written in 2.3. Procedures, the eligible people explained the study objectives to them, and asked them to read and sign an informed consent form if they wished to participate in the study. We used a written consent form.

5. "Assuming the correlation between the social support and fear and experience of childbirth, at least 0.16"author express reference?

Answer: Thank you for your comments. Yes, the reference is correct. Given the correlation coefficient of 0.16 between social support and experience of childbirth, and using the following formula and considering 15% loss, we determined the sample size.

6. Persian Version of the Pregnancy Worries and Stress Questionnaire (PWSQ)

7. This tools generally was used in assessing anxiety?

Answer: Thank you for your comments. No, this questionnaire assesses mothers' concern in six subcategories: mother's health, newborn’s health, experience of childbirth and motherhood, mother–newborn bonding, personal-family and personal-occupational. It does not assess anxiety.

8. "The second stage was performed after the delivery. At this stage, one week before the probable date of the delivery, the mother was contacted via a phone call and asked to inform the researcher when they attended the hospital. The researcher then visited the hospital to collect information on the mother's childbirth experience and perceived social support when she went to the postpartum ward and was in a more stable condition. If a mother's clinical condition were not suitable for any reason, the completion of the questionnaires would be postponed until the mother's condition stabilized". If women did 

Answer: Thank you for your comments. I did not quite understand your question. Do you mean that this process was done or not? If so, I should say yes. One of the researchers did it and set the time and mothers collaborated with her.

9. Which kind of social support (instrumental support-emotional – information- appraisal) had the highest negative causal relationship with fear in pregnant women?

Answer: Thank you for your comments. The mentioned domains are related to General social support but, as explained before, we used Multidimensional Scale of Perceived Social Support (MSPSS), it measures support from three sources, including the family, friends, and significant others with 12 items on a seven-point Likert scale, from strongly disagree to agree strongly. Perceived social support is the individual's evaluation of the availability of support when necessary and required and is a qualitative-mental and measurable concept. The total score of MSPSS is the sum of the scores of all its items (3) and we use its total score in path analysis. 

Reviewer #2: 

1. The first aim of the current study is to examine the effect of spiritual well-being, perceived social support and pregnancy's worries and stress on fear of childbirth. However, little information about pregnancy's worries and stress can be found in the Introduction part. 

Answer: Thank you for your comments. We added some papers in the Introduction section as per reviewer comments.

Furthermore, what kind of variable is pregnancy's worries and stress? An independent variable of fear of childbirth? Or a mediating variable between fear of childbirth and childbirth experience? In Figure 2, the path analysis revealed a potential mediation model among PE, TEN, and EX. Therefore, from my humble point, the results of the study do not fit the aims.

Answer: Thank you for your comments. Pregnancy worries and stress (TEN) is endogenous for variables of mother’s education, father’s education, gravidity, number of children, socioeconomic status, perceived social support, spiritual health, and fear of childbirth and exogenous for childbirth experience. Endogenous variables are variables that are diagrammed as being influenced by other variables in the model. The variables diagrammed as independent of any influence are the exogenous variables. Dependent variables are always endogenous, but some independent (or predictor) variables can be endogenous if they are themselves being influenced by other independent variables in the model (1). Fear of childbirth is endogenous for all variables except TEN, and endogenous for TEN. 

It is correct that TEN is a mediator between childbirth experience and fear of childbirth. It was one of our aims, but we did not write it in our questions in the end of the introduction. So, we added a questions at the end of Introduction: “Do pregnancy worries and stress mediate the effect of fear of childbirth on childbirth experience?. 

The book ‘MUNRO’S Statistical Methods for Health Care Research’ (2013) explained that, in general, the research questions for path analysis relate to the testing of relationships that are hypothesized to exist between and within a dependent variable and a set of predictor variables. In general, path analysis helps us address the following questions: 

1. Are the paths in the model supported by the data?

2. What is the total effect (direct plus indirect) of a predictor variable?

3. Does one of the independent variables mediate the effect of another variable on the dependent variable? 

Path analysis is literally an analysis of the paths or lines in a model that represent the influence of one variable on another. It is used to answer research questions about the effect of a given independent (X1) variable on the dependent variable (Y) in the model and path analysis is an extension of conventional regression that shows not only the direct effects but also the indirect effects of each variable on the dependent variables, and the results can be used to provide a rational interpretation of the relationships and correlations observed (1) , so because we assessed direct and indirect effects of all variables as well as pregnancy worries and stress (TEN), we considered mediator effect on childbirth fear and experience in the present study. All path coefficient show in Table 3.

. The authors should explain clearly the relationships of the five variables (spiritual well-being, perceived social support, pregnancy's worries and stress, fear of childbirth, and childbirth experience) in the Introduction part.

Answer: Thank you for your comments. We added some papers as per reviewer comments:

“Numerous factors, such as demographic, psychosocial, and spiritual characteristics, are associated with adverse experiences and fear of childbirth [13]. Also, according to the Fear-Tension-Pain theory, fear of childbirth, maternal tension, and the amount of pain experienced are cyclically related and can affect each [14]. In addition, spiritual health and perceived social support can help as a coping mechanism to control oneself in difficult and stressful situations and show better adaptation [16,17]. Even if the results of various studies show the positive effect of spirituality on mental health,but we have to considered that Religious beliefs may bring In case of negative results by encouraging to quit or Get out of treatment, without timely referral to prevent diseases[16]

2. From the Introduction part, perceived social support might be an independent variable of fear of childbirth. However, in Method part, fear of childbirth was measured in the first stage, while perceived social support was measured in the second stage. It is very confusing and the authors should explain it.

Answer: Thank you for your comments. In our manscript, the aim was to assess perceived social support during pregnancy until delivery and its effect on chilbirth experience too, not only after delivery or childbirth fear. So we assessed it after delivery. According to the litreature, social relations, the unofficial support networks of pregnant women, and strong support by midwives can strengthen women's perception of childbirth as a physiological and controllable processes (4). Social support is also a predictor of postpartum depression and maternal role competence. So it has to start before pregnancy until after delivery (5). 

Furthermore, in our study, we wanted to assess childbirth experience. Childbirth experience and fear of childbirth mutually affect each other, so it was important to assess each variable first and remove its confounding effect on the other variables because fear of childbirth is an important issue during pregnancy and during and after childbirth. (4) For omiting this interventional effect, our team decided to assess fear of childbirth in the first phase (last 4 weeks) and childbirth experience in the secound phase (after delivery).

3. The minimum required sample size (352) is equal to actual sample size (352)?

Answer: Thank you for your comments. The researchers tried to collect all of the sample size determined.

4. There are some obvious limitations in the study. However, the Limitation part is short and non-professional. The authors should re-write this part.

Answer: Thank you for your comments. It was added as per reviewer comments.

5. The Conclusion part is less related to the findings of the study. For example, countries have always considered mothers' health as one of the most important groups in society. One of the most important health indicators of a country is its maternal mortality rate. Is it the conclusion from the results of the current study? The authors should re-write this part and conclude what we can get from the findings of the current study.

Answer: Thank you for your comments. We corrected it as per reviewer comments.

6. In Table 3, pregnancy worries and stress was included in the model of childbirth experience but excluded in the model of fear of childbirth. It is not consistent with the aims of the study. The authors should explain the reason.

Answer: Thank you for your comments. Sorry for the mistake. It was added.

7. The format of tables is not standard, such as Table 1. Moreover, the format of fonts is inconsistent, such as Table 1 and Table 2. The authors should uniform it.

Answer: Thank you for your comments. We corrected all.

Regards

Researchers Team

1. Plichta SB, Kelvin EA, Munro BH. Munro's statistical methods for health care research: Wolters Kluwer Health/Lippincott Williams & Wilkins; 2013.

2. Sharif M, Zaidi A, Waqas A, Malik A, Hagaman A, Maselko J, et al. Psychometric validation of the Multidimensional scale of perceived social support during pregnancy in rural Pakistan. Frontiers in Psychology. 2021;12:601563.

3. Zimet GD, Dahlem NW, Zimet SG, Farley GK. The multidimensional scale of perceived social support. Journal of personality assessment. 1988;52(1):30-41.

4. Fisher C, Hauck Y, Fenwick J. How social context impacts on women's fears of childbirth: a Western Australian example. Social science & medicine. 2006;63(1):64-75.

5. Saeieh SE, Rahimzadeh M, Yazdkhasti M, Torkashvand S. Perceived social support and maternal competence in primipara women during pregnancy and after childbirth. International journal of community based nursing and midwifery. 2017;5(4):408.

---

## [Decision Letter · Decision Letter 1]

15 Aug 2023

PONE-D-23-08753R1The relationships of spiritual health, pregnancy worries and stress and perceived social support with childbirth fear and experience: A path analysisPLOS ONE

Dear Dr. Mahmoodi,

Thank you for submitting your manuscript to PLOS ONE. After careful consideration, we feel that it has merit but does not fully meet PLOS ONE’s publication criteria as it currently stands. Therefore, we invite you to submit a revised version of the manuscript that addresses the points raised during the review process.

We look forward to receiving your revised manuscript.

Kind regards,

Zemenu Yohannes Kassa

Academic Editor

PLOS ONE

Reviewers' comments:

Reviewer's Responses to Questions

**Comments to the Author**

1. If the authors have adequately addressed your comments raised in a previous round of review and you feel that this manuscript is now acceptable for publication, you may indicate that here to bypass the “Comments to the Author” section, enter your conflict of interest statement in the “Confidential to Editor” section, and submit your "Accept" recommendation.

Reviewer #2: (No Response)

Reviewer #3: All comments have been addressed

2. Is the manuscript technically sound, and do the data support the conclusions?

Reviewer #2: Partly

Reviewer #3: Partly

3. Has the statistical analysis been performed appropriately and rigorously? 

Reviewer #2: Yes

Reviewer #3: Yes

4. Have the authors made all data underlying the findings in their manuscript fully available?

Reviewer #2: Yes

Reviewer #3: Yes

5. Is the manuscript presented in an intelligible fashion and written in standard English?

Reviewer #2: No

Reviewer #3: Yes

6. Review Comments to the Author

Reviewer #2: 1. There are some conflicting statements in the manuscript. For example, in Limitation section, authors said “Another limitation was the cross-sectional nature of the study that deprives the researcher from the opportunity to follow up ideas.” However, in Methods section, this longitudinal (prospective) study was performed in 2021 on eligible pregnant mothers presenting to the selected health centers in Qazvin, Iran. What is the kind of this study design actually? Besides, in Limitation section, part of the study was done after delivery, which may have tired mothers and affected their responses. But in Methods section, if a mother's clinical condition were not suitable for any reason, the completion of the questionnaires would be postponed until the mother's condition stabilized. Limitation section is very important for the current study and future research. Authors should take more efforts to improve it.

2. The Conclusion section can hardly cover main findings of this study. It is very simple and unspecific. For example, in Conclusion section, our study showed that spiritual health, and perceived social support is related to childbirth fear and experience in pregnant mothers. However, if authors want to get this finding, it is not necessary to conduct a path analysis. On the other hand, the third aim of this study is to explore what is the childbirth experience (direct/indirect) on fear of childbirth in pregnant mothers? The fifth aim is to explore if pregnancy worries and stress mediate the effect of fear of childbirth on childbirth experience? These research questions are not reflected well in the subsequent sections, such as Results section, Discussion section, and Conclusion section. In other words, these sections seem like they come from different researches and different authors. Authors should try to make different sections of the manuscript become closely related.

Reviewer #3: Dear Editor,

I want to thank you for providing the opportunity to review this revised manuscript, ““The relationships of spiritual health, pregnancy worries and stress and perceived social support with childbirth fear and experience: A path analysis?”

Fear of childbirth is defined as the fear experienced before, during and after birth. While fear of childbirth affects the mother, fetus and newborn in many ways, it also causes negativities in the relations between parents. It is important to define the fear of childbirth by the health personnel, to determine the factors causing the fear and to plan the appropriate interventions. For this reason, the results of the related study will make significant contributions to the existing literature.

My evaluation notes:

The manuscript includes a current and very interesting subject. Therefore, I congratulate the authors

Referee suggestions were carried out significantly on the text. However, there are a couple of points that are not fully understood.

1. What is the total number of pregnant women followed in the center where the study was conducted?

2. It would be appropriate to indicate the dates of the study and the duration of the study.

3. Were there any cases not included or excluded from the study? How many and why?

4. Women who have fear of childbirth are often affected by their “delivery choice”. It is stated that the fear of childbirth increases the rate of cesarean section and that a large part of women prefer cesarean delivery instead of vaginal delivery. Was the mode of delivery examined in this study?

5. Were the pregnancies of the pregnant women included in the study planned and desired pregnancies?

6. Another factor that causes fear of birth is previous negative birth and abortion experiences. Was it evaluated in this study?

7. How the study was carried out is not fully understood. Were the cases included in the study evaluated twice, before and after birth? Were these interviews conducted face-to-face or over the phone? Were the scales and questionnaires given to the subjects and asked to be filled in at home? In this case, erroneous results may occur.

8. It is recommended to expand the discussion section with the results of studies conducted in different cultures on the subject. The following studies can be used.

Chválna Zuzana, Dominová Natália, Ostatníková Michaela, et al. Prevalence and risk factors for serious birth concerns in unselected population of mothers. Prevalencia a rizikové faktory závažných obáv z pôrodu u neselektovanej populácií rodičiek. Ceska Gynekol. 2023;88(2):80-85. doi:10.48095/cccg202380

Nieminen K, Stephansson O, Ryding EL. Women's fear of childbirth and preference for cesarean section--a cross-sectional study at various stages of pregnancy in Sweden. Acta Obstet Gynecol Scand. 2009;88(7):807-813. doi:10.1080/00016340902998436

Viirman F, Hesselman S, Wikström AK, et al. Negative childbirth experience - what matters most? a register-based study of risk factors in three time periods during pregnancy. Sex Reprod Healthc. 2022;34:100779. doi:10.1016/j.srhc.2022.100779

Räisänen S, Lehto SM, Nielsen HS, Gissler M, Kramer MR, Heinonen S. Fear of childbirth in nulliparous and multiparous women: a population-based analysis of all singleton births in Finland in 1997-2010. BJOG. 2014;121(8):965-970. doi:10.1111/1471-0528.12599

Smith KB, Zdaniuk B, Ramachandran SO, Brotto LA. A longitudinal case-control analysis of pain symptoms, fear of childbirth, and psychological well-being during pregnancy and postpartum among individuals with vulvodynia. Midwifery. 2022;114:103467. doi:10.1016/j.midw.2022.103467

9. It seems that the study was conducted during the period when the most important effects of COVID-19 were seen. The cases participating in the study may have been affected by this situation. It can be stated in the limitations of the study.

10. Do the cases participating in the study have a previous history of psychiatric disorder and treatment?

11. Superiority and dissimilarity of this study from the other studies have not been emphasized enough. What is the difference should be emphasized

12 .What is the recommendation for further studies?

In my opinion, this study is publishable after minor revision in your journal.

Best regards,

7. PLOS authors have the option to publish the peer review history of their article (what does this mean?). If published, this will include your full peer review and any attached files.

Reviewer #2: No

Reviewer #3: No

---

## [Author Response · Author response to Decision Letter 1]

27 Aug 2023

Dear Editor 

Thank you for your valuable comments. We corrected and answerd all of them as follows:

Reviewer #2: 1. There are some conflicting statements in the manuscript. For example, in Limitation section, authors said “Another limitation was the cross-sectional nature of the study that deprives the researcher from the opportunity to follow up ideas.” However, in Methods section, this longitudinal (prospective) study was performed in 2021 on eligible pregnant mothers presenting to the selected health centers in Qazvin, Iran. What is the kind of this study design actually? Besides, in Limitation section, part of the study was done after delivery, which may have tired mothers and affected their responses. But in Methods section, if a mother's clinical condition were not suitable for any reason, the completion of the questionnaires would be postponed until the mother's condition stabilized. Limitation section is very important for the current study and future research. Authors should take more efforts to improve it.

Answer: Thank you for your attention. This study was longitudinal and we corrected. We rewrite limitation part, and highlighted. 

2. The Conclusion section can hardly cover main findings of this study. It is very simple and unspecific. For example, in Conclusion section, our study showed that spiritual health, and perceived social support is related to childbirth fear and experience in pregnant mothers. However, if authors want to get this finding, it is not necessary to conduct a path analysis. On the other hand, the third aim of this study is to explore what is the childbirth experience (direct/indirect) on fear of childbirth in pregnant mothers? The fifth aim is to explore if pregnancy worries and stress mediate the effect of fear of childbirth on childbirth experience? These research questions are not reflected well in the subsequent sections, such as Results section, Discussion section, and Conclusion section. In other words, these sections seem like they come from different researches and different authors. Authors should try to make different sections of the manuscript become closely related.

Answer: Thank you for your attention. If we only wanted to determine the relationship between this variables, yes it wasn’t need to use path analysis, but, we wanted to understand the direct effect and indirect effect of these variables, so we had to do the powerful analysis like path analysis.About third and fifth questions, we added them in result ,discussion and conclusion parts and highlighted. 

Thanks 

Reviewer #3: Dear Editor,

1. What is the total number of pregnant women followed in the center where the study was conducted? 

Answer: Thank you for your attention. In this study Two Referral Center of Qazvin were selected) Kowsar and shafa hospitals) and 352 eligible pregnant mothers were assessed. We added it in methods part and highlighted.

2. It would be appropriate to indicate the dates of the study and the duration of the study.

Answer: Thank you for your comment. This study was conducted from 2021/3/3 to 2021/9/2 .we added in methods part and highlighted. 

3. Were there any cases not included or excluded from the study? How many and why?

Answer: Thank you for your comment. We didn’t have any case that not included or excluded from study.

4. Women who have fear of childbirth are often affected by their “delivery choice”. It is stated that the fear of childbirth increases the rate of cesarean section and that a large part of women prefer cesarean delivery instead of vaginal delivery. Was the mode of delivery examined in this study? 

Answer: Thank you for your comment.in this study we assessed all variable in path analysis. Path analysis requires the same type of data as linear multiple regression. In other words, you need a dependent variable that is continuous and normally distributed. ( Munro's statistical methods for health care research: Wolters Kluwer Health/Lippincott Williams & Wilkins; 2013) Accordingly, we couldn't assess the mode of delivery. Because of that, we only entered the mothers who wanted to have a vaginal delivery and did it. 

5. Were the pregnancies of the pregnant women included in the study planned and desired pregnancies?

Answer: Thank you for your comment. Yes, all pregnancies were planned and desired, and for this subject, first, we asked women about their pregnancy, and if it was wanted, they entered the study. We added it to the inclusion criteria, and highlighted.

6. Another factor that causes fear of birth is previous negative birth and abortion experiences. Was it evaluated in this study?

Thank you for your comment. Yes, it was our inclusion criteria.

7. How the study was carried out is not fully understood. Were the cases included in the study evaluated twice, before and after birth? Were these interviews conducted face-to-face or over the phone? Were the scales and questionnaires given to the subjects and asked to be filled in at home? In this case, erroneous results may occur.

Answer: Thank you for your comment. Yes, as we wrote in the 2.3. Procedures: Data were collected in two stages in this study; the first stage was performed during the last four weeks of the pregnancy, and the second stage was performed after the delivery. The questionnaires were given to the participants to complete; if they could not complete the questionnaires in one meeting, the researcher would set the next time to complete the questionnaires, which means they came back to the center and completed, so all questionnaires were completed face-to-face and not at home.

8. It is recommended to expand the discussion section with the results of studies conducted in different cultures on the subject. The following studies can be used.

Chválna Zuzana, Dominová Natália, Ostatníková Michaela, et al. Prevalence and risk factors for serious birth concerns in unselected population of mothers. Prevalencia a rizikové faktory závažných obáv z pôrodu u neselektovanej populácií rodičiek. Ceska Gynekol. 2023;88(2):80-85. doi:10.48095/cccg202380

Nieminen K, Stephansson O, Ryding EL. Women's fear of childbirth and preference for cesarean section--a cross-sectional study at various stages of pregnancy in Sweden. Acta Obstet Gynecol Scand. 2009;88(7):807-813. doi:10.1080/00016340902998436

Viirman F, Hesselman S, Wikström AK, et al. Negative childbirth experience - what matters most? a register-based study of risk factors in three time periods during pregnancy. Sex Reprod Healthc. 2022;34:100779. doi:10.1016/j.srhc.2022.100779

Räisänen S, Lehto SM, Nielsen HS, Gissler M, Kramer MR, Heinonen S. Fear of childbirth in nulliparous and multiparous women: a population-based analysis of all singleton births in Finland in 1997-2010. BJOG. 2014;121(8):965-970. doi:10.1111/1471-0528.12599

Smith KB, Zdaniuk B, Ramachandran SO, Brotto LA. A longitudinal case-control analysis of pain symptoms, fear of childbirth, and psychological well-being during pregnancy and postpartum among individuals with vulvodynia. Midwifery. 2022;114:103467. doi:10.1016/j.midw.2022.103467

Answer: Thank you for your comment; we added and highlighted two references per your recommendation in the Discussion part.

9. It seems that the study was conducted during the period when the most important effects of COVID-19 were seen. The cases participating in the study may have been affected by this situation. It can be stated in the limitations of the study.

Answer: Thank you for your comment; we added it in the limitation part, and highlighted

10. Do the cases participating in the study have a previous history of psychiatric disorder and treatment?

Answer: Thank you for your comment. No. they didn’t have. This was our inclusion criteria as we wrote in inclusion criteria part:” not having a mental illness according to the self-reports or health records, and not taking antidepressants and anti-anxiety medications according to the self-reports or health records.”

11. Superiority and dissimilarity of this study from the other studies have not been emphasized enough. What is the difference should be emphasized

Answer: Thank you for your comment. As we wrote in the end of background:” To the best of the authors’ knowledge, perceived social support, spiritual and psychological variables, and fear and experience of childbirth have not been addressed yet in a single model.” It means this model can be first model that show casual paths between these variables.

12 .What is the recommendation for further studies?

Answer: Thank you for your comment. We wrote in the limitation part: “This study assessed pregnant mothers presenting to the selected health centers. Mothers who did not refer to the health center or offered in other specific centers did not set so that it could be another limitation. We recommended that in further research, they consider.” and also “The study was conducted during COVID-19 so the participating cases may have been affected by this situation” so ut can be done in another time.

Regards

Researchers Team

---

## [Decision Letter · Decision Letter 2]

26 Sep 2023

PONE-D-23-08753R2The relationships of spiritual health, pregnancy worries and stress and perceived social support with childbirth fear and experience: A path analysisPLOS ONE

Dear Dr. Mahmoodi,

Thank you for submitting your manuscript to PLOS ONE. After careful consideration, we feel that it has merit but does not fully meet PLOS ONE’s publication criteria as it currently stands. Therefore, we invite you to submit a revised version of the manuscript that addresses the points raised during the review process.

We look forward to receiving your revised manuscript.

Kind regards,

Zemenu Yohannes Kassa, Msc

Academic Editor

PLOS ONE

Journal Requirements:

Reviewers' comments:

Reviewer's Responses to Questions

**Comments to the Author**

1. If the authors have adequately addressed your comments raised in a previous round of review and you feel that this manuscript is now acceptable for publication, you may indicate that here to bypass the “Comments to the Author” section, enter your conflict of interest statement in the “Confidential to Editor” section, and submit your "Accept" recommendation.

Reviewer #2: All comments have been addressed

Reviewer #3: (No Response)

2. Is the manuscript technically sound, and do the data support the conclusions?

Reviewer #2: Yes

Reviewer #3: Yes

3. Has the statistical analysis been performed appropriately and rigorously? 

Reviewer #2: Yes

Reviewer #3: Yes

4. Have the authors made all data underlying the findings in their manuscript fully available?

Reviewer #2: Yes

Reviewer #3: Yes

5. Is the manuscript presented in an intelligible fashion and written in standard English?

Reviewer #2: Yes

Reviewer #3: Yes

6. Review Comments to the Author

Reviewer #2: 1. Limitation section is very important for the current study and future research. However, the Limitation section in the manuscript is still not professional and specific. For example, “we recommended that in further research, they consider.” Consider what? How to consider? Besides, “The study was conducted during COVID-19 so the participating cases may have been affected by this situation.” Affect what? Reporting bias? Or sampling procedure? How to handle it in future research? Authors should pay more attention to improve scientific writing quality of the whole manuscript.

Reviewer #3: Dear Editor,

I want to thank you for providing the opportunity to review this revised chapter ‘The relationships of spiritual health, pregnancy worries and stress and perceived social support with childbirth fear and experience: A path analysis”

I have re-reviewed the study. I determined that the authors made the suggested changes to the text.

In my opinion, this study is publishable in your journal.

Best regards,

7. PLOS authors have the option to publish the peer review history of their article (what does this mean?). If published, this will include your full peer review and any attached files.

Reviewer #2: No

Reviewer #3: No

---

## [Author Response · Author response to Decision Letter 2]

15 Oct 2023

Answer: Thank you. We checked and corrected it

---

## [Editor Report · Decision Letter 3]

13 Nov 2023

The relationships of spiritual health, pregnancy worries and stress and perceived social support with childbirth fear and experience: A path analysis

PONE-D-23-08753R3

Dear Dr. Zohreh Mahmoodi,

We’re pleased to inform you that your manuscript has been judged scientifically suitable for publication and will be formally accepted for publication once it meets all outstanding technical requirements.

Kind regards,

Zemenu Yohannes Kassa

Academic Editor

PLOS ONE

Additional Editor Comments (optional):

Authors should check punctuation for example, put full stop after reference citation.

Revised the following sentences, as follows Qazvin is the administrative city of the province, I think in Iran there is one capital city, which is Tehran.

Qazvin is the largest city and the capital of Qazvin Province in the central part of Iran.

.......experience. (Figure 3) (Table 3). please put full stop after figure or table.
---

## [Editor Report · Acceptance letter]

29 Nov 2023

PONE-D-23-08753R3 

The relationships of spiritual health, pregnancy worries and stress and perceived social support with childbirth fear and experience: A path analysis 

Dear Dr. Mahmoodi:

I'm pleased to inform you that your manuscript has been deemed suitable for publication in PLOS ONE. Congratulations! Your manuscript is now with our production department. 

Kind regards, 

on behalf of

Dr. Zemenu Yohannes Kassa 

Academic Editor

PLOS ONE